# Analysis of the Properties of Anticorrosion Systems Used for Structural Component Protection in Truck Trailers

**DOI:** 10.3390/ma17246303

**Published:** 2024-12-23

**Authors:** Wojciech Skotnicki, Dariusz Jędrzejczyk

**Affiliations:** Faculty of Mechanical Engineering and Computer Science, University of Bielsko-Biala, Willowa 2, 43-309 Bielsko-Biala, Poland; djedrzejczyk@ubb.edu.pl

**Keywords:** protective coatings, corrosion resistance, surface preparation, technical dry friction

## Abstract

The article compares the properties of coatings (cataphoretic, hot-dip zinc, and thermo-diffusion zinc) applied to steel components used in the automotive industry. The research focused on the analysis of corrosion resistance, hardness measurements, and tribological properties conducted on steel guides used in trailer and truck body structures as well as fasteners (M12 × 40 bolts). The base surfaces were cleaned chemically. Corrosion resistance was tested in a salt chamber, while coating thickness was measured using the magnetic induction method. Coating hardness (HV 0.02) was assessed with a microhardness tester, and tribological properties were tested under dry friction conditions. The results showed that the zinc coatings demonstrated corrosion resistance far superior to paint coatings.

## 1. Introduction

Corrosion is a major challenge in the durability of steel structures and automotive components, leading to significant material losses and operational failures. While a variety of protective coatings have been developed to combat this issue, discrepancies in the reported performance of different coating systems complicate the selection process [1,2,3,4,5,6,7]. This study aims to address this gap by systematically evaluating the mechanical and corrosion resistance properties of cataphoretic, hot-dip zinc, and thermo-diffusion zinc coatings applied to automotive steel components. In the automotive industry, components such as profiles and bolts are constructed from a wide range of metallic materials, including carbon steel, alloyed steel, and stainless and corrosion-resistant steels, as well as aluminum, magnesium, and titanium alloys [8,9]. There are many methods of preventing corrosion or protecting metals against corrosion, including the use of appropriate coatings that create an insulating barrier on the surface [10,11,12] and the use of corrosion inhibitors [13] or cathodic protection [14,15]. In order to extend the durability of vehicles, many protective coatings for steel have been developed, including chemical treatment and painting systems combined with the optimization of the shape of car profiles, as well as the use of additional treatments such as sealing and waxing [16]. The pressure to reduce production costs means that structural elements are increasingly made of cheaper materials that only guarantee appropriate mechanical properties—anti-corrosion properties are improved by additional processing—the application of a coating. Moreover, corrosion-resistant coatings have two basic advantages: they are effective, and the process of applying them is relatively simple [12]. Although there are protective coatings with so-called “intelligent materials” that have self-healing properties [17], and this effect can also be achieved through the use of shape memory polymers [18], environmental and economic factors seem to favor traditionally used metal and paint coatings. Zinc belongs to the most cost-effective elements traditionally employed in the fabrication of anti-corrosion coatings (Zn, Cu, Ni, and Cr) [19]. Furthermore, the deposition processes of Zn coatings are characterized by simplicity and very low financial investments [20]. According to the International Lead and Zinc Study Group report [21], global zinc production averaged approximately 12 million tons annually (years 2016–2020), with over 50% of this output being utilized for steel corrosion protection through galvanization processes. Zinc coatings are applied using four methods in various applications: hot-dip galvanizing, electrogalvanizing, zinc flake deposition and sherardizing (thermal diffusion) [22,23,24]. There is no consensus in the literature concerning the corrosion resistance of various coatings [25,26,27,28,29]. Hot-dip galvanizing is the most widely used method for protecting industrial steel parts [25,30,31]. This process provides high-quality, long-term corrosion resistance, making it applicable across various environments, starting from marine and rural to industrial settings, as well as in diverse industries. The corrosion protection offered by hot-dip galvanizing typically lasts up to 75 years, depending on the coating thickness and the specific environmental exposure to corrosive agents [30]. Fasteners are mainly protected through hot-dip zinc galvanizing [23], which typically results in coating thicknesses ranging from 45 to 65 μm [4]. The microstructure of these coatings has a characteristic diffusive nature and comprises several intermetallic phases of the Fe–Zn system [32]. A coating with a similar structure, composed of intermetallic phases, is also achieved through thermo-diffusion galvanizing [26]. On the other hand, comparative accelerated tests conducted in a neutral salt spray environment revealed that hot dip zinc coatings (HD) exhibit a markedly higher susceptibility to corrosion compared to thermo-diffusion (TD) coatings [25]. Specifically, the tests indicated that the thermo-diffusion coating provided corrosion protection for up to 1000 h, whereas the hot-dip zinc coating only lasted for 250 h—under identical conditions. Thermo-diffusion coatings enhance the corrosion resistance of high-strength steel used in bridge wire applications. However, under specific conditions, these coatings can provide superior corrosion resistance and extended service life compared to conventional hot-dip galvanizing treatments [26]. It is important to note that thermo-diffusion galvanized steel sheets exhibited lower corrosion resistance in comparison to hot-dip galvanized steel sheets [27]. The electroplated bolts showed the lowest corrosion resistance in the salt spray environment in relation to the thermal diffusion and hot-dip coating [28]. It was observed that the galvanized bolts showed “red corrosion” on the entire surface (test time—1500 h in NSS spray). In contrast, the hot-dip galvanized bolts exhibited corrosion primarily affecting the nut and a part of the threaded section, while the thermo-diffusion galvanized bolts demonstrated only minimal signs of iron corrosion. The prevailing opinion in the literature suggests that both the mechanical and anticorrosion properties of diffusion zinc layers exceed those of other zinc coatings that do not undergo diffusion processes [29]. Sometimes the corrosion resistance of a coating may depend on its microstructure [33]. For example—the corrosion resistance of zinc lamella coatings is influenced by the size of the flakes; flake size reduction contributes to increasing the anti-corrosion properties of the coating. Research conducted by other scientists [34] indicates that the properties of lamellar coatings also depend on the epoxy resin content within the layer, the optimal anti-corrosion protection values are observed in the range of 30–35% resin. In comparison, the corrosion resistance of cataphoretic coatings in a salt spray environment is reported to range between 480 and 1000 h [35]. The corrosion rate of galvanized vehicle components is significantly influenced by climatic conditions [16]. Notably, severe corrosion has been reported in Europe, whereas corrosion rates in areas of warmer climate were observed to be less severe than those in snowy regions. This difference can be attributed to various environmental factors. Specifically, although high temperatures and relative humidity are prevalent in Southeast Asia, these conditions may not always lead to more aggressive corrosion. In fact, the combination of heat and moisture can sometimes act as a barrier, slowing down the corrosion process. On the other hand, in colder, drier climates, like those in Europe, the use of de-icing salts during winter can create conditions that accelerate corrosion. Therefore, despite the higher temperatures in tropical regions, the presence of other factors, such as salt and condensation in colder climates, can make snowy regions more prone to severe corrosion. Since the second half of the 20th century, cataphoretic coatings have been extensively utilized in the automotive industry to reduce suspension corrosion [36]. The cataphoretic painting process has many advantages: the potential for easy automation and stable coating thickness with minimal deviations, even in hard-to-reach areas that are unsuitable for powder coating. This technique does not produce streaks, and the surface coverage efficiency is notably high, with small material losses. Furthermore, the emission of volatile organic compounds into the atmosphere is significantly limited. Moreover, the geometry of the coated element does not affect the cataphoretic coating corrosion resistance [36]. A notable disadvantage of cataphoretic painting is the requirement for the need to make specialized drainage holes in the painted structures—antennas are often positioned on vehicle roofs not only to enhance signal reception but also to cover up one of the drainage holes [35]. Additionally, the primary threat to the durability of graphite-gray coatings is ultraviolet radiation. Tests conducted in accordance with the PN-EN ISO 16474-2 standard [37] revealed that the cataphoretic coating exhibited a loss of functional properties in a relatively short period of time (<100 h of exposure). Suspension elements that are not exposed to direct sunlight do not require additional processing [35]. Accurately assessing corrosion risks is a critical process in the effective design of anti-corrosion protection systems. The PN-EN ISO 12944-2 standard [38] identifies the most corrosive environments as follows: aggressive industrial settings, marine areas, and major urban thoroughfares during peak traffic hours. The corrosion rate is influenced by several factors, including humidity, temperature, and pollutant levels (chemical composition). Various chemical compounds can alter water from moisture into an electrolyte, thereby promoting the formation of corrosion sites [39,40]. The anti-corrosion protection mechanism of paint coatings is very complex and depends on many factors: the interaction of anti-corrosion pigments, the integrity of the coating, and the adhesion to the substrate surface [41]. Among the various stages of anti-corrosion protection technology, the most critical and challenging step is the preparation of the surface prior to the deposition of the coating [42,43,44,45,46]. The coated surface is not always thoroughly cleaned and contains various substances: including reaction products from the surrounding environment (such as scale and rust), as well as fats, oils, greases, and atmospheric deposits (such as dust and water-soluble salts). The removal of these contaminants generally necessitates multiple cleaning operations. The quality of bolted connections and their durability are influenced by many factors resulting from: the adopted solution (type of material, type of heat treatment, lubrication method, and adopted tolerances); the specificity of the connected parts (type of coating and roughness); the type of working environment (corrosion rate); the characteristics of the tightening process (tightening method, tightening speed and method of controlling the tightening force) [47]. The friction coefficient µ is one of the most important parameters (strictly defined during the design of joints), which affects the quality of the joint at every stage of use: during assembly, work in the joint (self-locking), or disassembly. Too high values of the coefficient may result in damage resulting from incomplete tightening or even breaking of the bolt, while too low values of the friction coefficient (<0.08) may result in automatic loosening of bolt connections. Currently, the friction coefficient value in fasteners typically required by the automotive industry is in the range of 0.10 to 0.19 (±0.03). In the engineering industry, the required friction coefficient values are higher and range from 0.21 to 0.24. According to the data presented in the publication [48], the commonly used protective coatings can provide friction coefficient values in a fairly wide range of 0.04 to >0.30.

The above literature analysis shows that there are many discrepancies regarding the properties of zinc coatings. The conducted review demonstrates that in some cases the zinc coating obtained through hot-dip galvanizing exhibits greater corrosion resistance, which is considered as the one of the crucial parameters used for coating quality assessment. Additionally, in many cases, the authors highlight the corrosion resistance of the thermo-diffusion coating as the most significant. A key factor in this comparison is the testing of real structural parts and complete anti-corrosion systems, especially when different coatings are used for protection.

Based on the literature data presented above, which indicates comparable costs of zinc and KTL coatings (dependent on protection duration and the dimensions of the protected elements [47,48]), as well as similar corrosion resistance between the coatings under comparison [23,25,26,27,28,29,30,31,32,33,34,35,36,37], this study focuses on examining both mechanical and performance properties of the chosen coatings (cataphoretic coating, hot dip zinc, and thermo-diffusion zinc), including corrosion resistance tests and friction coefficient measurements. The obtained results were compared with findings from the authors’ previous research on zinc coatings [46,49,50,51] conducted based on laboratory samples taken from the research material. In the discussed work, the studies were carried out on actual components (set of parts: steel guide—mounting bolts) protected against corrosion in industrial conditions.

## 2. Materials and Methods

The paper presents results of tests of structural parts used in the construction of flatbed semi-trailers with a tarpaulin cover. The tests were performed on guides made of S355MC steel (Figure 1) used in the construction of trailers and truck bodies. The guide consists of two cold-rolled elements: a guide plate and a bracket. The elements preventing the guide from moving vertically were M12 × 40 bolts made of 23MnB4 steel.

The chemical composition analysis of carbon and sulfur was conducted using the Leco CS844 analyzer (LECO Corporation, St. Joseph, MI, USA). For the remaining elements, the chemical composition was examined with the iCAP Q mass spectrometer. Before being submitted for testing, the surfaces of the samples were mechanically cleaned and chemically degreased. The dimensions of the samples were 10 × 10 mm. The chemical composition of the materials of the tested structural parts is presented in Table 1.

The guides were cataphoretically painted—the cleaned element was immersed in a bath of water-soluble CathoGuard 900 paint with simultaneous flow of electric current. The main parameters influencing the coating thickness were: the value of the current voltage and the electrolyte deposition time. The bolts used to fasten the guide were thermal diffusion and hot-dip galvanized in industrial conditions. The parameters of the conducted processes are listed in Table 2.

The study includes the measurement of the following parameters:Surface roughness was measured using the Phase View system (PhaseView, Verrières-le-Buisson, France) with ZeeScan software version 2.4, employing a non-contact microscopic attachment method;The chemical composition analysis was conducted using: carbon and sulfur analyzer LECO CS844 (LECO Corporation, St. Joseph, MI, USA), mass spectrometer by: Thermo Fisher Scientific (iCAP Q, Waltham, MA, USA);The phase analysis using SEM was performed on the JEOL JSM-7800F scanning electron microscope. (JEOL Ltd. Akishima, Tokyo, Japan);Coating thickness was assessed using a PosiTector 6000MP magnetic induction tester with a 90° depth finder (DeFelsko, New York, NY, USA);Hardness measurements were measured at the cross-section of the tested coatings and the subsurface layer of steel using a Vickers hardness tester (HV 0.02) with a Mitutoyo Micro-Vickers HM-210 A device (Model 810–401 D, Mitutoyo, Kawasaki, Japan);Coating corrosion resistance was evaluated using an Ascott CC2000 salt chamber (Ascott, Staffordshire, UK) according to the PN-EN ISO 9227:2017-06 standard [54]. The tests utilized a corrosive medium of NaCl at a concentration of 50 g/dm^3^, with a solution density of 1.035 g/cm^3^ and a fall value of 1.033 g/cm^3^, pH 6.7, air pressure of 1 bar, and a chamber temperature of 40 °C. Post-testing, the samples were cleaned in a 15% hydrochloric acid (HCl) solution containing 1% corrosion inhibitor PICKLANE 60 (COVENTYA, Weiland, Germany).Tribological properties tests were carried out using the T-11 tester produced by ITEE, Radom, Poland.

## 3. Results and Discussion

### 3.1. Microscopic Observations

The preparation of samples for microscopic observations and hardness measurements involved:cutting and preliminary shaping of the samples;hot mounting;grinding using water-resistant papers of various grit sizes,polishing with diamond suspensions.

The collected samples for testing measured 10 × 10 mm. Microscopic observations, chemical composition analysis, and hardness measurements were performed on the cross-section of the sample taken from the guide (arrow—Figure 1) as well as on the cross-section of the bolt head.

The functional properties of the protective coating are a function of such parameters as: adhesion, hardness, flexibility, self-locking characteristics, wear resistance, ultraviolet (UV) stability, and impact resistance [55]. One of the basic factors influencing the above properties is the microstructure of the coating. Figure 2 shows selected results of microscopic examinations of the tested coatings’ cross-sections.

Microscopic analysis was supplemented by EDS studies, which allowed for verification of the range of occurrence of individual layers visible in the tested zinc coatings. The range of occurrence of layers was marked in Figure 2a–c using colored markers. According to the Fe–Zn system [24,25,26], three phases occur in the hot-dip/immersion coating: Γ(Fe_3_Zn_10_), δ(FeZn_10_, FeZn_7_), ζ(FeZn_13_)—and solid solution of iron in zinc—η. The darker lines running perpendicular to the steel surface visible in layer δ (Figure 2b,d) are not cracks but natural elements of the structure of this phase. The zinc coating deposited by thermal diffusion (sherardization) exhibits characteristics comparable to those of hot-dip coatings but does not include the η phase, as illustrated in Figure 2a [48,49,50]. Although there are some discrepancies in the literature regarding the structure of this coating, the prevailing view is that it is composed of two phases: Γ and δ.

The cataphoretic coating presented in Figure 2e was mostly homogeneous and compact, but occasional cracks or porosities could be observed. The thickness of the protective coatings measured using the magnetic induction method is presented in Table 3.

### 3.2. Hardness and Roughness Tests

During the tests, 10 measurements were performed on each sample. Analysis of the obtained results presented in Figure 3 indicates that the highest hardness on the surface is characteristic of the thermo-diffusion coating.

In the outer layer of the thermo-diffusion coating, values close to 370 HV 0.02 (±6 HV 0.02) were measured, while the hardness of the coating near the steel surface reached a value of approximately 340 HV 0.02 (±5 HV 0.02). A decreasing trend in the change of hardness was observed over the entire cross-section of the thermo-diffusion coating. The changes shown are a consequence of changes in the microstructure observed in the cross-section of the coating (Figure 2). The EDS analysis confirmed that the analyzed coating consists of two layers (Γ+δ), which are formed as a result of iron diffusion into the coating during the galvanizing process. The research conducted by Pokorny [56] shows that the δ phase is about 10% harder than the Γ phase—the hardness measured in the δ phase even reached values ranging from 330 to 460 HV. According to data from the publication [16,52], the δ phase is about 15% harder than the Γ phase. As a result of the increased hardness of the zinc coating, its wear during friction is reduced [57] moreover, the transition from the η phase to the ζ and δ phases causes a decrease in the friction coefficient [50] and thus a decrease in the self-locking of the bolted connection.

The hardness of the hot-dip zinc coating varies from about 50 HV 0.02 near the outer surface to about 300 HV 0.02 near the surface of the galvanized steel. As in the case of the thermo-diffusion coating, the change in hardness is determined by the type of microstructure and the properties of the individual phases (η—Zn–70 HB; ζ—FeZn_13_–220HB; δ—FeZn_7_–270 HB; and Γ_1_—Fe_5_Zn_21_–350 HB [23,52,53]).

The hardness of the cataphoretic coating is an order of magnitude lower than the hardness of zinc coatings. The cathodic electroplating coating is homogeneous and does not exhibit such a layered structure as zinc coatings, therefore the hardness determined on its cross-section (approx. 45 HV 0.02) does not change significantly, and the observed small variation is caused by the occurrence of a few cracks and pores.

The roughness of the tested coatings depends on the method of their production. The comparison of the measured values is presented in Figure 4. Zinc coatings deposited at an elevated temperature are characterized by roughness (Ra) in the range of 2.0–2.3 μm, while the roughness of the cataphoretic coating does not exceed 1 μm.

Figure 5 presents an example of a 2D profile and surface topography of a hot-dip zinc coating.

### 3.3. Friction Coefficient Measurement

The next stage was testing of friction characteristics, which was carried out using the T-11 tester. Tribological tests were conducted on the surfaces of samples taken from the research object. For the guide, a 50 × 50 mm sample was taken, while for the bolt connection, the test was carried out on the upper surface of the bolt head. The rotation speed of the sample was 30 rpm. During the test, at a unit load of 0.25 MPa, the friction force and temperature were continuously recorded, which was in the range of 22–32 °C. The counter sample used for testing was made of 23MnB4 steel and had a diameter of 4 mm. Figure 6 shows the changes in the friction coefficient during a single measurement of the tested coatings.

The friction coefficient for the hot-dip galvanized coating shows a rapid increase in the initial stage of the test (from 250 to 400 s), reaching a value of approximately 0.4. After that, the rate of increase clearly slows down, and the coefficient stabilizes at around 0.55 after about 1900 s. This behavior may be due to gradual wear of the surface coating and the exposure of a rougher zinc structure. During the friction test, the zinc material may wear off, revealing harder layers, which increases the resistance to movement.

In the case of the thermo-diffusion coating, the friction coefficient quickly stabilizes at a value of approximately 0.43. This stability suggests that the thermo-diffusion coating has a uniform structure and high wear resistance under friction conditions. The thermo-diffusion process ensures a strong bond between the protective layer and the substrate, which reduces coating wear and provides predictable tribological properties. It can be assumed that the microhardness of this coating and its resistance to plastic deformation play a key role in stabilizing the friction coefficient.

The friction coefficient for the cataphoretic coating remains low and constant in the initial stage of the test (up to 600 s), at approximately 0.1, after which it increases gradually and stabilizes at around 0.32 after approximately 1500 s. The initially low friction coefficient could be due to the presence of a smooth, homogeneous surface of the cataphoretic coating, which reduces the resistance to movement. The gradual increase in the coefficient is likely the result of the coating’s wear and the exposure of the substrate with different mechanical properties. This phenomenon suggests that the cataphoretic coating is less resistant to wear compared to the other coatings, leading to a gradual increase in friction as the coating wears.

Figure 7 compares the average values of the friction coefficient from 6 measurements.

The course of changes in the recorded characteristics corresponds well with the microstructure of individual coatings, their properties, and wear expressed as mass loss (Δm) and depth of the worn groove (d). The wear of the hot-dip, thermal diffusion, and cataphoretic coatings reached the following values: Δm_HD_ = 0.03 g; Δm_T_ = 0.006 g; Δm_C_ = 0.004 g; d_O_ = 25 μm; d_T_ = 5 μm; and d_K_ = 35 μm. The cross section of the tested coatings in the place of tribological wear is shown in the Figure 8.

However, the cataphoretic coating was completely worn out—as a result of the test the mass loss values should be considered taking into account that the specific gravity of the zinc coating is approximately 7× greater than that of the cataphoretic coating. The appearance of samples after the tribological test is shown in Figure 9.

The hardness of the hot-dip zinc coating increases as the coating wear progresses deeper. The outer soft phase η (approx. 55 HV 0.02) was probably partially worn out during the first stage of the test—contact formation, and the gradual increase in the friction coefficient value in the subsequent stages of the test results from the contact with the much harder phases ζ and δ (250–300 HV 0.02). The thermo-diffusion coating is in turn composed of a mixture of the hard phases δ and Γ. The δ phase has the highest hardness among the analyzed phases (380 HV 0.02)—this may reduce the rate of coating chipping.

### 3.4. Corrosion Resistance Test

The final stage of the investigation involved assessing the corrosion resistance of the tested components in the Ascott CC2000 salt chamber. The test was conducted in accordance with the PN-EN ISO 9227:2017-06 standard [54], utilizing three samples from each group: bolts with hot-dip and thermo-diffusion zinc coatings, as well as guides with a cataphoretic coating. An example of the appearance of the tested components following the corrosion test is presented in Figure 10.

The parameters for the accelerated corrosion tests conducted in the salt chamber (NSS) were as follows: corrosive medium of NaCl at a concentration of 50 g/dm^3^, solution density of 1.035 g/cm^3^, precipitation value of 1.033 g/cm^3^, pH of the solution at 6.7, air pressure maintained at 1 bar, and an internal chamber temperature of 40 °C. Upon removal from the chamber, the tested components were cleaned using a 15% aqueous hydrochloric acid (HCl) solution supplemented with 1% corrosion inhibitor (Zinc Phosphate). The surface conditions of the samples were evaluated at 24 h intervals in accordance with the standard. The criterion for corrosion resistance was defined as the duration until the onset of “red corrosion” on the surfaces of the tested elements.

Among the tested coatings, the hot-dip coating showed the highest corrosion resistance—840 h. On the bolts with the thermo-diffusion coating, red corrosion appeared after 48 h, while on the guides with the cataphoretic coating, red corrosion centers were visible after 72 h. The significantly lower corrosion resistance of the zinc thermal diffusion coating may result from the microstructure of its outer layer, which is usually more cracked and porous than the hot-dip zinc coating, where the outer layer is a compact and homogeneous solid solution of iron in zinc (η). In this case, even the cataphoretic coating proved to be more resistant to corrosion than thermo-diffusion, which is probably due to the smaller number of pores and cracks visible in the cross-section of the coating.

Corrosion of protective coatings is an electrochemical process that involves both anodic and cathodic reactions. In the case of zinc coatings, the anodic reaction results in the oxidation of zinc (Zn → Zn^2+^ + 2e^−^), while the cathodic reaction involves the reduction of water to hydrogen (2H_2_O + 2e^−^ → H_2_ + 2OH^−^). Thermodiffusion coatings, due to their porosity and cracks, may promote the formation of galvanic cells, where the areas not covered by the coating act as the anode, while the rest of the surface behaves as the cathode. These micro-scale galvanic cells can accelerate the corrosion process. In contrast, hot-dip coatings exhibit a more homogeneous microstructure, which may explain their superior corrosion resistance.

## 4. Conclusions

Based on a detailed analysis of the available literature on the subject and the results obtained during the conducted research, the following conclusions have been formulated:The research conducted indicates that the type of applied zinc coating should be closely dependent on the operating conditions. In cases where a more aggressive corrosive environment is present, the best results are achieved with a hot-dip galvanized coating. In conditions of higher exposure to abrasive wear, better results are obtained with a thermo-diffusion zinc coating. A paint coating is generally sufficient for moderate exposure to corrosion and abrasive wear, especially where aesthetic considerations are key.The differentiation of the microstructure of the tested coatings has a direct impact on the measured characteristics of the friction coefficient changes and the average value of this parameter. The highest friction coefficient value was determined for the hot-dip zinc coating μ_HD_ = 0.4, a slightly lower value was measured for the thermo-diffusion coating μ_T_ = 0.3, and the lowest for the cataphoretic coating—μ_C_ = 0.19. When selecting a protective coating for a bolts connection, the self-locking of the thread should also be taken into consideration, which can vary widely depending on the kind of coating used.During the tribological test, the greatest wear was observed in the case of the cataphoretic coating, where the depth of the worn groove was equal to the total thickness of the coating (35 μm). In the case of the thermo-diffusion coating, the greatest abrasion resistance was achieved—the groove depth was only 5 μm, while the zinc hot-dip coating was worn to a depth of 25 μm.In terms of measured corrosion resistance, the most effective protection for automotive components is provided by a hot-dip zinc coating. This coating significantly outperforms other protective coatings in terms of durability. The corrosion resistance of hot-dip zinc, as quantified by the duration until the onset of red corrosion, is more than ten times greater than that of alternative coatings. This remarkable performance is attributed to the zinc’s ability to form a robust, sacrificial layer that effectively shields the underlying metal from environmental factors such as moisture, salts, and oxygen. As a result, components coated with hot-dip zinc experience a much slower degradation process, leading to extended service life and reduced maintenance costs. This makes it an ideal choice for automotive applications, where long-term durability and reliability are crucial.Future research on the discussed coatings should focus on evaluating their performance under varied environmental conditions, including: UV radiation, cyclic humidity, temperature fluctuations, and exposure to industrial pollutants. Further studies could also investigate the long-term durability and adhesion of the coatings under mechanical stress and repeated loading.

## Figures and Tables

**Figure 1 materials-17-06303-f001:**
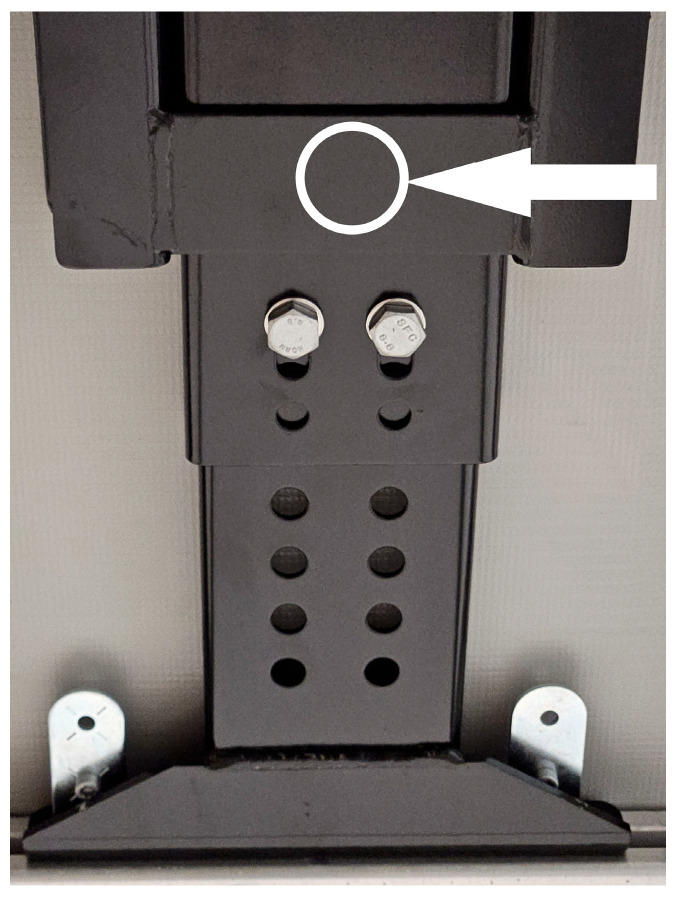
Steel guide appearance with the applied bolt connection (the location where samples were taken for testing is marked with a white arrow).

**Figure 2 materials-17-06303-f002:**
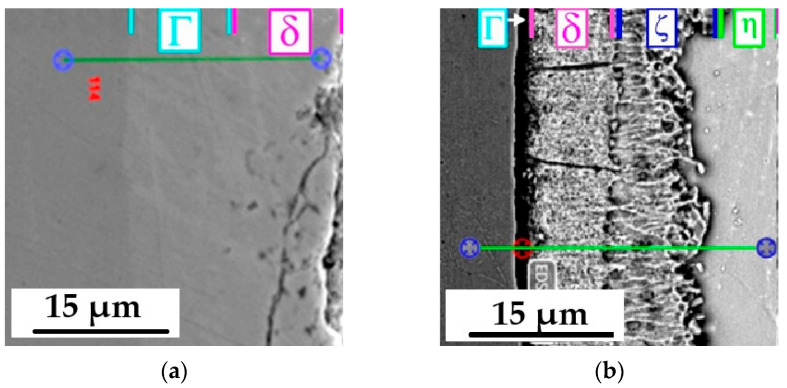
Microstructure observed on the cross-section of the tested coatings using a scanning microscope: (**a**)—thermo-diffusion zinc coating, (**b**,**d**)—hot-dip zinc coating, (**c**)—EDS analysis of thermo-diffusion coating’s cross section according to the green line, (**e**)—cataphoretic coating.

**Figure 3 materials-17-06303-f003:**
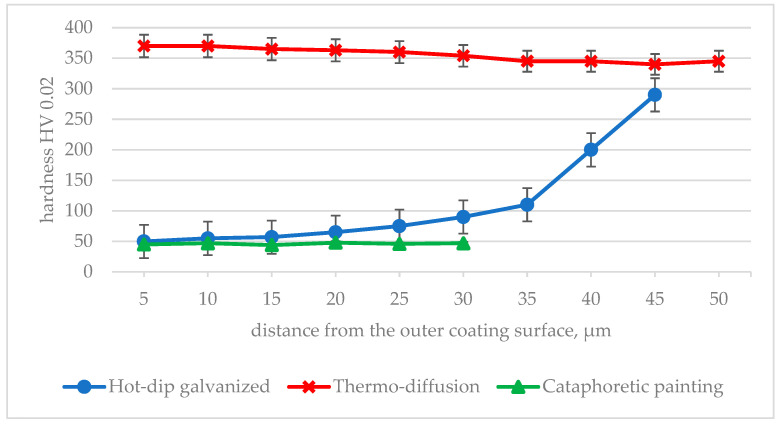
Change in hardness measured on the cross-section of the tested coatings.

**Figure 4 materials-17-06303-f004:**
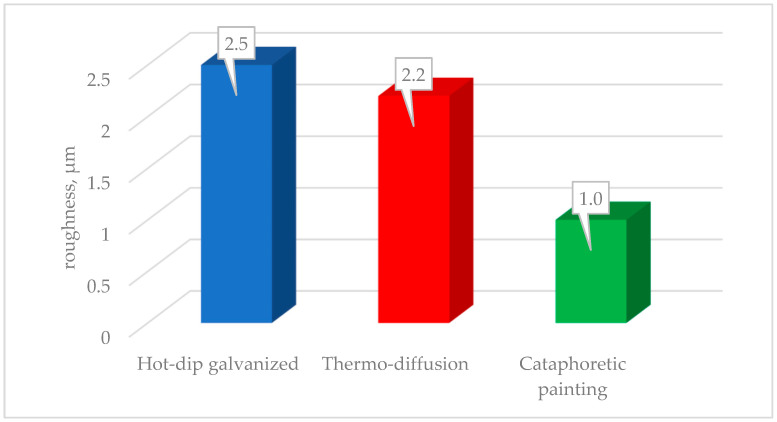
Comparison of the roughness of the tested coatings.

**Figure 5 materials-17-06303-f005:**
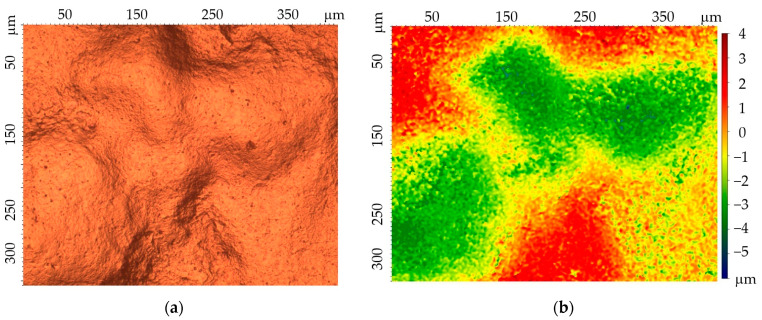
An example of a roughness profile (**a**) and surface topography of a hot-dip zinc coating (**b**).

**Figure 6 materials-17-06303-f006:**
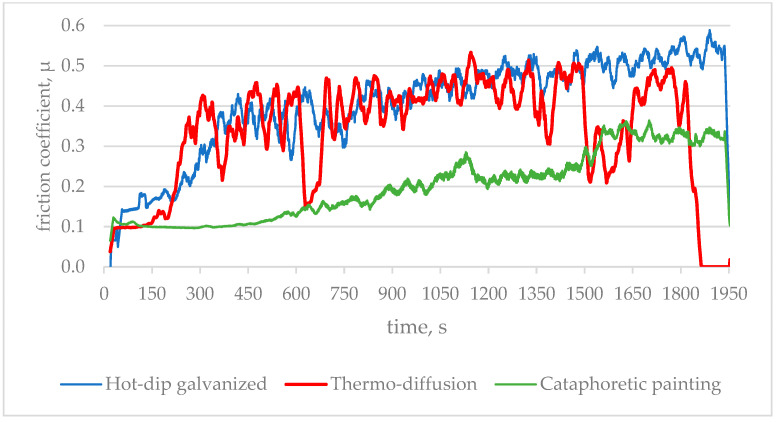
The course of changes in the friction coefficient value during a single measurement.

**Figure 7 materials-17-06303-f007:**
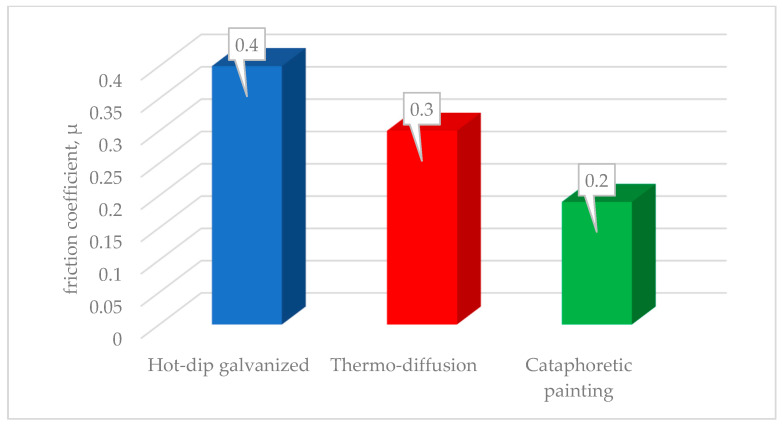
Comparison of the average friction coefficient values determined for the tested coatings.

**Figure 8 materials-17-06303-f008:**
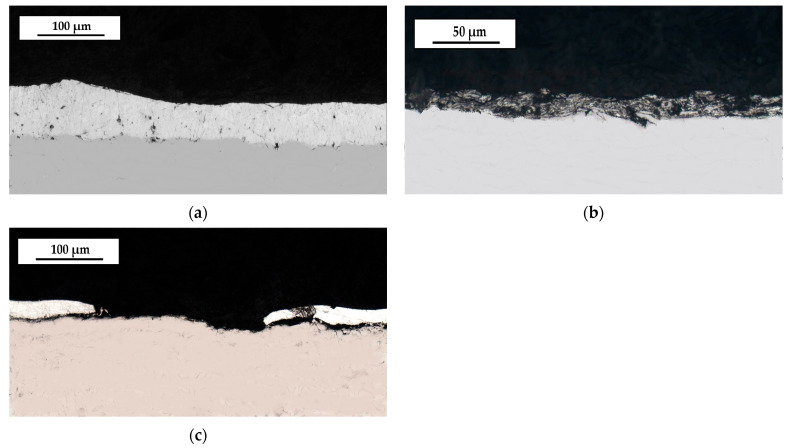
The cross-section microstructure of the coating after tribological tests: (**a**)—hot-dip coating, (**b**)—thermo-diffusion coating, (**c**)—cataphoretic coating.

**Figure 9 materials-17-06303-f009:**
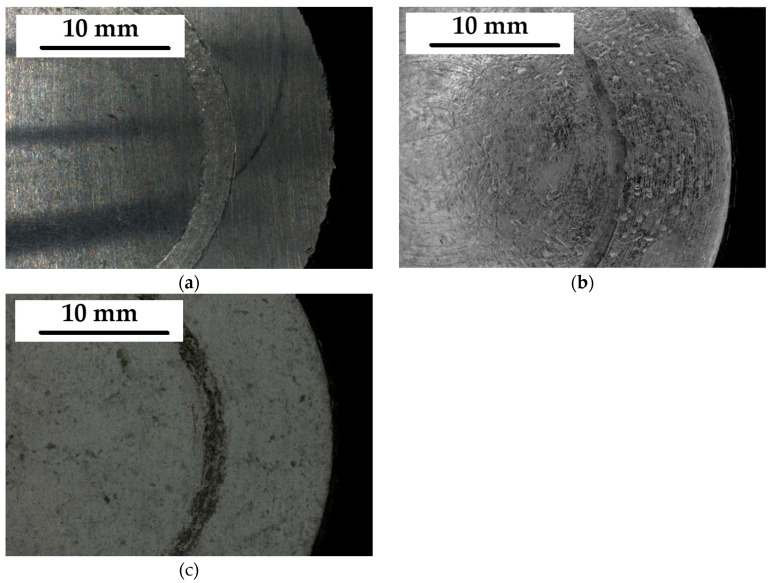
Sample surface after tribological test: (**a**)—hot-dip coating, (**b**)—thermodiffusion coating, (**c**)—cataphoretic coating.

**Figure 10 materials-17-06303-f010:**
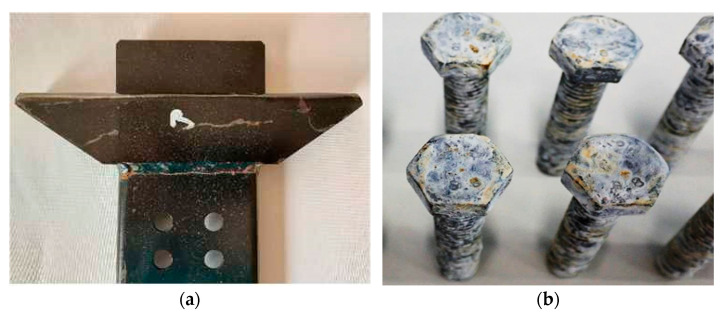
Tested elements after the corrosion test: (**a**)—guide; (**b**)—M12 × 40 bolts.

**Table 1 materials-17-06303-t001:** Chemical composition of tested parts.

Chemical Composition of Steel, %.								
	C	Si	Mn	P	S	Cr	Cu	Ni
23MnB4 (1.5535)	0.2	0.1	1.1	0.02	0.02	0.3	0.2	0.005
	C	Si	Mn	P	S	Al	Nb	V
S355MC (1.0976)	0.1	0.01	1.0	0.02	0.01	0.005	0.05	0.1

**Table 2 materials-17-06303-t002:** Methodology of deposition of tested coatings.

Sample No.	Kind of Coating	Sample Preparation Methodology
1	Cataphoretic painting according to PN-EN ISO 12944-2:2018-02 [38]	The samples were subjected to etching in a 15% hydrochloric acid (HCl) solution, followed by immersion in CathoGuard 900, a water-soluble coating produced by BASF Coating AG, Münster, Germany (pH range: 5.5–7.0; voltage: 220–250 V; deposition rate 10–12 μm/min; duration: 250 s). The coated samples were then dried for 1 h at a temperature of 200 °C.
2	Hot-dip galvanized according to PN-EN ISO 10684:2006 [52]	Samples were etched in a 15% hydrochloric acid (HCl) solution, fluxed, and hot-dip galvanized at 480 °C in a zinc (Zn) bath with aluminum (Al 0.1%), bismuth (Bi 0.05%), and nickel (Ni 0.05%) additives, duration: 240 s, followed by water cooling.
3	Thermo-diffusion according to PN-EN ISO 17668 [53]	Samples were etched in a 15% hydrochloric acid (HCl) solution, followed by galvanization in a powder with filler and activator in a rotary chamber at 10–20 rpm, 425 °C for 5 h, and then cooled in air to 22 °C.

**Table 3 materials-17-06303-t003:** Coating thickness measurement results.

No. of Measurements and the Thickness of the Coating, µm.											
	1	2	3	4	5	6	7	8	9	10	Average (s.d)
Hot-dip galvanized	40 MIN	43	43	47 MAX	45	45	44	45	47	46	44.5 (2.1)
Thermo-diffusion	55 MAX	52	53	51	53	50 MIN	54	54	51	52	52.5 (1.7)
Cataphoretic painting	31 MIN	35 MAX	32	35	33	34	31	35	34	32	33.2 (1.6)

## Data Availability

The original contributions presented in this study are included in the article. Further inquiries can be directed to the corresponding author.

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
