# Peer review of "Analysis of the Properties of Anticorrosion Systems Used for Structural Component Protection in Truck Trailers"

_materials, 2024, doi:10.3390/ma17246303_

Round 1

Reviewer 1 Report (Previous Reviewer 1)

Comments and Suggestions for Authors
1. Figure 2 shows the phase distribution at the coating interface. The text mentions that the phase distribution was determined using EDS (Energy Dispersive Spectroscopy) element mapping. However, I have some concerns regarding Figure 2(a). The two phases shown in the image do not exhibit clear regional variations in the electron microscope image, whereas distinct phase regions can be observed in the other subfigures. I suggest including the EDS maps in the article to help readers distinguish the phases more easily.

The article mentions that the structure of the coating has a significant impact on its performance, but it does not provide macroscopic images of the coating cross-section 3.  In Figure 3, the hardness of the thermal diffusion coating is only tested up to 20 μm, while Table 3 shows that the average thickness of the coating is 33 μm. Since the hardness test does not reach the coating surface, could this overlook potential variations in surface hardness?

Comments on the Quality of English Language

For all the figures in the article, I recommend standardizing the font format and size for the text within the images.

Author Response

Dear Reviewer,

We sincerely thank you for preparing a detailed review and for the time you dedicated to thoroughly evaluating our work. We truly appreciate your engagement and valuable comments, which will allow us to further improve the manuscript. We are grateful for your constructive suggestions, which will undoubtedly contribute to enhancing the quality of our paper and making it better aligned with the expectations within the given research area.

We hope that the revisions will be satisfactory and that our work will gain recognition from both reviewers and readers.

Best regards,

The Authors

Reviewer 2 Report (New Reviewer)

Comments and Suggestions for Authors

This is a review of the paper entitled “Analysis of the Properties of Anticorrosion Systems used for 2 Structure Parts Protection in Truck Trailers” by Wojciech Skotnicki and Dariusz JÄ™drzejczyk submitted for Materials journal.

The paper investigates the performance of various protective coatings applied to steel components in the automotive industry. The study addresses a significant topic with practical implications. Several aspects of the manuscript require some refinement and I also have made some suggestions.

The abstract is too detailed in some areas. A more concise summary focusing on the key findings, methods, and implications could be written. I have a suggested revision: "The study investigates the performance of anti-corrosion coatings (cataphoretic, hot-dip zinc, and thermo-diffusion zinc) applied to steel components in automotive applications. Tests included corrosion resistance, hardness, and tribological properties under simulated conditions. Results show that hot-dip zinc coatings outperform others in corrosion resistance, while thermo-diffusion coatings excel in hardness and wear resistance. These findings guide the selection of optimal coatings for specific applications."

The transitions between sections are abrupt, and the introduction does not build a compelling narrative that links the research problem to the methods employed. For instance, the introduction does not sufficiently highlight the specific gaps this study aims to address. I would suggest to consider restructuring the introduction highlighting more the novelty and significance of the work. The introduction section lacks focus. It introduces numerous aspects of corrosion and protection without emphasizing the study's objectives. Some references are updated, and the relevance of certain details (e.g., global zinc production statistics) to the study's core focus is unclear. If I may suggest some improvement: condense the review of protective coatings and focus on knowledge gaps, such as discrepancies in reported performance among coating types, and link these directly to the study's aim, plus include recent references.

The Materials and Methods Section is detailed and replicable but the rationale for selecting specific testing parameters, such as temperature and pH for the salt chamber, should be explicitly stated. In this section I suggested you to consider including a summarizing table for the parameters for each test. Also, the biggest problem I noticed is the samples location. What samples and from where did you analyze? Sample preparation for microscopic observations is missing and should be provided in very detailed manner.

The results are well presented but you should include discussion comparing findings to previous studies. I would also evaluate the limitations of the experimental setup, such as differences between laboratory and real-world conditions, if any. My concise suggestion is to discuss possible causes for the observed variations in coating performance.

The figures and tables are generally clear but Table 1 should present the exact determined composition of the alloy (either alone or compared to the standard(included) for precise conformity). Figure 1. Remove a) and insert analyzed samples position. Figure 2: explain the lines and colors into a legend or directly into the figure caption. Figure 3: please provide a more professional image. Figure 4 – idem Figure 3 and: are the decimals trusted? (2.54 micron absolute measurement require a different equipment?). Figure 5. Measurement units for abscise axes is missing. There is too much white space between and below. The image 6 showing friction coefficient changes, require more contextual explanation and standard deviations and error margins should be addressed in the discussion. Figure 7 -idem Figure 3. I suggest you to modify the caption for each figure/table so that provides sufficient standalone information. Also please modify the font inside figures to the body text font type.

The conclusions summarize the findings but is to long and some paragraphs must be moved to the discussion section. Also, include a brief mention of limitations and areas for future research.

In the reference list some sources are outdated.

I would also rewrite some parts of the text. I suggest you to do it, at least in the introduction section, so I am providing a suggested change in the text

"A wide variety of materials are used to make structural elements, each requiring specific types of anti-corrosion protection. When designing steel structures and metal machine parts, it is crucial to consider the durability of materials in both natural and industrial environments. The damages related to corrosion are a leading cause of material losses, ranking ahead of environmental pollution and various disasters at sea, on roads, or in the air." can be replacd with "Corrosion is a major challenge in the durability of steel structures and automotive components, leading to significant material losses and operational failures. While a variety of protective coatings have been developed to combat this issue, discrepancies in the reported performance of different coating systems complicate the selection process. This study aims to address this gap by systematically evaluating the mechanical and corrosion resistance properties of cataphoretic, hot-dip zinc, and thermo-diffusion zinc coatings applied to automotive steel components."

Comments on the Quality of English Language

Good

Author Response

Dear Reviewer,

We sincerely thank you for preparing a detailed review and for the time you dedicated to thoroughly evaluating our work. We truly appreciate your engagement and valuable comments, which will allow us to further improve the manuscript. We are grateful for your constructive suggestions, which will undoubtedly contribute to enhancing the quality of our paper and making it better aligned with the expectations within the given research area.

We hope that the revisions will be satisfactory and that our work will gain recognition from both reviewers and readers.

Best regards,

The Authors

Reviewer 3 Report (New Reviewer)

Comments and Suggestions for Authors

The paper is original with minor issues to be addressed before to proceed with the publication:

Page 2 line94: raported = reported.

In the same line starts a sentence "Notably, ..." that shall be reconsidered because it is not clear and leads to conclusions not supported by evidences. Specifically, it is not clear why an environment "hot and wet" should be less aggressive than a cold and dry one. At the same time, in the same sentence the example starts from Europe to end in Asia without a logical sequence of facts.

Page 3 line 110: what is the acronym KTL?

Page 3 lines 149-150: the sentence "Also ... coating." is not clear

Page 4 line 178: "electrolyte deposition time" is it time or rate? because time without rate is quite meaningless to define the efficacy and to present the resulting coating thickness.

Page 5 Table 2: The nature of the coating shall be exposed. Sample No. 2 is not marked how long this coating lasts while this information is indicated for samples 1 and 3.

Page 5 line 191: was = were

Page 6 figure 2: Please mark the contrast used. fig (c) shows a gap between the metal substrate and the coating, the same gap is described as a phase.

Page 11 lines 322-339: there is the need of more "electrochemistry" to better describe the oxidation/corrosion phenomena. This part is missing and the conclusions are affected by that. Moreover, pictures like fig. 10 are interesting but missing of details that would increase the quality of the presentation of experimental results.

Page 12 Lines 341-366: the conclusions start immediately with a list of points. An introduction would be appreciated and is missing. Point 5 is too short and should be further developped.

Author Response

Dear Reviewer,

We sincerely thank you for preparing a detailed review and for the time you dedicated to thoroughly evaluating our work. We truly appreciate your engagement and valuable comments, which will allow us to further improve the manuscript. We are grateful for your constructive suggestions, which will undoubtedly contribute to enhancing the quality of our paper and making it better aligned with the expectations within the given research area.

We hope that the revisions will be satisfactory and that our work will gain recognition from both reviewers and readers.

Best regards,

The Authors

Round 2

Reviewer 2 Report (New Reviewer)

Comments and Suggestions for Authors

There are still some parts that need to be amended (compulsory). 

Figure 1: does not explain enough the analyzed samples areas (from WHERE was cut?? the exact area ....you must insert in the picture an arrow or something...). In the text you still have figure 1a and figure 1b. You didn't provide my request regarding samples preparation for the microscopy and hardness analyses. 

Table 1: The chemical composition of the materials of the tested structural parts is presented but how was determined (by which method? using what instrument? on what samples?? on what part of the samples? exactly where?).

For each method you must describe the procedure and the corresponding description of samples - the exact procedure and the exact samples (size, section, preparation, e.s.o.).

In table 3 something is wrong.

Figure 5 still not inserted the measurement units for the vertical axes.

Conclusions still bad. Look like results. No real conclusions dropped.

Author Response

Dear Reviewer,

We sincerely thank you for your time and for providing a detailed and constructive review of our manuscript. We greatly appreciate your valuable comments and suggestions, which will allow us to further improve our work. Your analysis and recommendations have contributed to enhancing the quality of our manuscript and better aligning it with the expectations in the given research field. We hope that the revisions will meet your approval and that our work will be positively received by other reviewers and readers.

Best regards,

The Authors

Round 3

Reviewer 2 Report (New Reviewer)

Comments and Suggestions for Authors

The submitted version is appropriate. 

Round 1

Reviewer 1 Report

Comments and Suggestions for Authors

This study compares the properties of zinc coatings prepared through cataphoretic painting, thermal diffusion, and hot-dip galvanizing. However, the influence of different preparation methods on the microstructure of these coatings, as well as the role of microstructure in corrosion resistance, has not been thoroughly investigated. The specific review comments are as follows:

1)Introduction section: The logic is unclear, and the advantages of zinc coatings, along with the characteristics and research progress of different preparation methods, are not well-highlighted. Additionally, it emphasizes the superiority of thermo-diffusion zinc coating over other methods, such as hot-dip galvanizing. Thus, the novelty and significance of the present work need clarification.

2) Line 155, Page 4:the bolts used to fasten the guide were thermal diffusion and hot-dip galvanized in industrial conditions. The coating prepared by cataphoretic painting is missing.

3)Fig. 2: Large interfacial cracks are clearly visible in the hot-dip and cataphoretic coatings. Does this impact the integrity and mechanical properties of these coatings?

4)Line 193, Page 6:The zinc coating deposited by thermal diffusion (sherardization) exhibits characteristics comparable to those of hot-dip coatings. However, the microstructure of these coatings differs significantly.

5)Line 303, Page 11:Among the tested coatings, the hot-dip coating showed the highest corrosion resistance – 840 h”. This statement contradicts the description in the Introduction. What is the underlying mechanism for the higher corrosion resistance of the hot-dip coating compared to other coatings?

Comments on the Quality of English Language

Ensure font consistency in all figures. Add spaces between numbers and units in the labels.

Author Response

(The authors gave the same response as above.)

Reviewer 2 Report

Comments and Suggestions for Authors

This is an interesting paper. However, it must be amended according to the following points:

Page 1.- Abstract section.- Lines 19 and 20 read: It was demonstrated that the tested zinc 19 coatings showed corrosion resistance many time greater then paint coating……it should read: It was demonstrated that the tested zinc 19 coatings showed corrosion resistance many times greater than paint coating.

Page 4.- Table 2.- Regarding Sample No. 2 preparation methodology, please indicate the wt% of Al, Bi and Ni additives in the Zn bath.

Page 5.- Line 168. For the hardness measurements reported, a Micro-Vickers HM-210 A tester was used. However, this equipment does not have nano-hardening capabilities. So, please delete the words and nano in the Abstract section (page 1), i.e., line 16 read: using a micro and nano hardness tester....it should read: using a microhardness tester.

Pages 6 and 8.- Undoubtedly, the micrographs in Figures 2 and 8 were obtained using a SEM/EDS equipment. Please include in Section 2.- Materials and Methods, all information regarding this equipment (model, manufacturer, city, country)

Page 8.- Figure 6.- On the individual friction coefficient data for the thermo-diffusion coating, significant drops in this coefficient are observed in time ranges of approx. 600-780 s and 1500-1750 s. Could the authors explain this behavior?

Page 9.- Figure 7.- Inside Fig. 7, the blue bar caption reads Hot-dip halvanized….Should read: Hot-dip galvanized.

Page 9.- Lines 264-267 read: The course of changes in the recorded characteristics corresponds well with the microstructure of individual coatings, their properties and wear expressed as mass loss (Δm) and depth of the worn groove (d). The wear of the hot-dip, thermal diffusion and cataphoretic coatings reached the following values: ΔmHD=0.03 g; ΔmT=0.006 g; ΔmC=0.004 g; General comment/question.- The hot-dip zinc coating showed the highest corrosion resistance after exposure in the salt fog chamber. However, the reverse was true when exposed to the tribological tests, where the hot-dip zinc coating disclosed the highest mass loss compared to the other coating systems.

In a scenario where the different coating systems used in the present work would be exposed to erosion-corrosion, which of the proposed systems could perform better? Could the authors comment on this? 

 Page 11.- Line 299 read: ….solution supplemented with 1% corrosion inhibitor. Comment: Please state the type or name of the inhibitor used.

Page 12.- References section: Include the DOI information in the following references: 12, 27, 31, 37, 41, 53, 54.

Page 12.- Reference 5 read: Fatigue of Aircraft Structure….Should read: Fatigue of Aircraft Structures

Page 13.- Reference 36: indicate city, state and country of the mentioned Conference

Page 14.- Reference 50, lines 464-464 read: Construction and Design 463 2, 82-91, 2011……Should read: Construction and Design 2011, 463 2, 82-91, doi…..

Author Response

(The authors gave the same response as above.)

Reviewer 3 Report

Comments and Suggestions for Authors

Authors talk about the Performance Properties of Selected Protective Coatings Applied on Steel Parts of Automotive Structures. It is well written manuscript with good research potential and practical application. 

Author Response

(The authors gave the same response as above.)

Reviewer 4 Report

Comments and Suggestions for Authors

Authors have discussed about the zinc coating development for steel surfaces. This coating helps prevent surface corrosion.

1. Authors have discussed about the advantage of using zinc coating over conventional coatings. Authors can consider eloborationg and comparing different types of coating with zinc coating. This will help authors understand why zinc coating is still preferred over conventional coatings. Howother coating performance compare with same? 

2. Authors can also consider doing electrochemical impedance spectroscopy (EIS) to support their conclusion.

3. Authors have evaluated the performance of coating using salt spray technique. Authors can also considering adding accelerated weathering testing to further evaluate performance.

Author Response

(The authors gave the same response as above.)
